# WildGaussians: 3D Gaussian Splatting in the Wild

**Jonas Kulhanek**[1,2,3,][*] **Songyou Peng**[3,][†] **Zuzana Kukelova**[4], **Marc Pollefeys**[3], **Torsten Sattler**[1]

[1] Czech Institute of Informatics, Robotics and Cybernetics, Czech Technical University in Prague
[2] Faculty of Electrical Engineering, Czech Technical University in Prague
[3] Department of Computer Science, ETH Zurich
[4] Visual Recognition Group, Faculty of Electrical Engineering, Czech Technical University in Prague

https://wild-gaussians.github.io

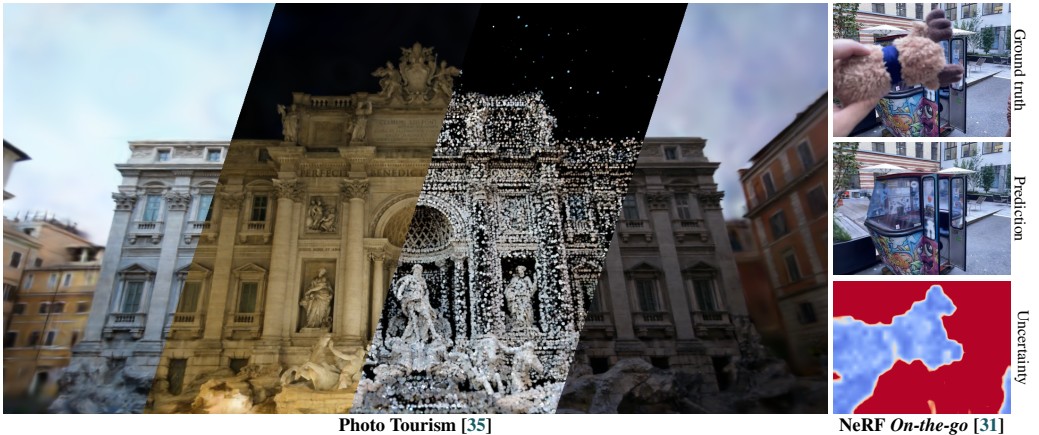

Figure 1: **WildGaussians** extends 3DGS [14] to scenes with appearance and illumination changes **(left)**. It jointly optimizes a DINO-based [27] uncertainty predictor to handle occlusions **(right)**.

## Abstract

While the field of 3D scene reconstruction is dominated by NeRFs due to their photorealistic quality, 3D Gaussian Splatting (3DGS) has recently emerged, offering similar quality with real-time rendering speeds. However, both methods primarily excel with well-controlled 3D scenes, while in-the-wild data – characterized by occlusions, dynamic objects, and varying illumination – remains challenging. NeRFs can adapt to such conditions easily through per-image embedding vectors, but 3DGS struggles due to its explicit representation and lack of shared parameters. To address this, we introduce WildGaussians, a novel approach to handle occlusions and appearance changes with 3DGS. By leveraging robust DINO features and integrating an appearance modeling module within 3DGS, our method achieves state-of-the-art results. We demonstrate that WildGaussians matches the real-time rendering speed of 3DGS while surpassing both 3DGS and NeRF baselines in handling in-the-wild data, all within a simple architectural framework.

## 1 Introduction

Reconstruction of photorealistic 3D representations from a set of images has significant applications across various domains, including the generation of immersive VR experiences, 3D content creation

---

[*]The work was done during an academic visit to ETH Zurich.
[†]Corresponding author, now at Google DeepMind.

38th Conference on Neural Information Processing Systems (NeurIPS 2024).

for online platforms, games, and movies, and 3D environment simulation for robotics. The primary objective is to achieve a multi-view consistent 3D scene representation from a set of input images with known camera poses, enabling photorealistic rendering from novel viewpoints.

Recently, Neural Radiance Fields (NeRFs) [1, 25, 37, 30, 38, 26, 9, 17, 29] have addressed this challenge by learning a radiance field, which combines a density field and a viewing-direction-dependent color field. These fields are rendered using volumetric rendering [12]. Despite producing highly realistic renderings, NeRFs require evaluating numerous samples from the field per pixel to accurately approximate the volumetric integral. Gaussian Splatting (3DGS) [14, 50, 49, 51, 15, 54] has emerged as a faster alternative. 3DGS explicitly represents the scene as a set of 3D Gaussians, which enables real-time rendering via rasterization at a rendering quality comparable to NeRFs.

Learning scene representations from training views alone introduces an ambiguity between geometry and view-dependent effects. Both NeRFs and 3DGS are designed to learn consistent geometry while simulating non-Lambertian effects, resolving ambiguity through implicit biases in the representation. This works well in controlled settings with consistent illumination and minimal occlusion but typically fails under varying conditions and larger levels of occlusion. However, in practical applications, images are captured without control over the environment. Examples include crowd-sourced 3D reconstructions [34, 1], where images are collected at different times, seasons, and exposure levels, and reconstructions that keep 3D models up-to-date via regular image recapturing. Besides environmental condition changes, e.g., day-night changes, such images normally contain occluders, e.g., pedestrians and cars, with which we need to deal with during the reconstruction process.

NeRF-based approaches handle appearance changes by conditioning the MLP that presents the radiance field on an appearance embedding capturing specific image appearances [24, 38, 24]. This enables them to learn a class of multi-view consistent 3D representations, conditioned on the embedding. However, this approach does not extend well to explicit representations such as 3DGS [14], which store the colors of geometric primitives explicitly. Adding an MLP conditioned on an appearance embedding would slow down rendering, as each frame would require evaluating the MLP for all Gaussians. For occlusion handling, NeRFs [24, 31] use uncertainty modeling to discount losses from challenging pixels. However, in cases with both appearance changes and occlusions, these losses are not robust, often incorrectly focusing on regions with difficult-to-capture appearances instead of focusing on the occluders. While NeRFs can recover from early mistakes due to parameter sharing, 3DGS, with its faster training and engineered primitive growth and pruning process, cannot, as an incorrect training signal can lead to irreversibly removing parts of the geometry.

To address the issues, we propose to enhance Gaussians with trainable appearance embeddings and using a small MLP to integrate image and appearance embeddings to predict an affine transformation of the base color. This MLP is required only during training or when capturing the appearance of a new image. After this phase, the appearance can be "baked" back into the standard 3DGS formulation, ensuring fast rendering while maintaining the editability and flexibility of the 3DGS representation [14]. For robust occlusion handling, we introduce an uncertainty predictor with a loss based on DINO features [27], effectively eliminating occluders during training despite appearance changes.

Our contributions can be summarized as: (1) **Appearance Modeling:** Extending 3DGS [14] with a per-Gaussian trainable embedding vector coupled with a tone-mapping MLP, enabling the rendered image to be conditioned on a specific input image's embedding. This extension preserves rendering speed and maintains compatibility with 3DGS [14]. (2) **Uncertainty Optimization:** Introducing an uncertainty optimization scheme robust to appearance changes, which does not disrupt the gradient statistics used in adaptive density control. This scheme leverages the cosine similarity of DINO v2 [27] features between training and predicted images to create an uncertainty mask, effectively removing the influence of occluders during training. The source code, model checkpoints, and video comparisons are available at: https://wild-gaussians.github.io/

## 2   Related work

**Novel View Synthesis in Dynamic Scenes.** Recent methods in novel view synthesis [25, 1, 14, 50] predominantly focus on reconstructing static environments. However, dynamic components usually occur in real-world scenarios, posing challenges for these methods. One line of work tries to model both static and dynamic components from a video sequence [19, 28, 43, 44, 10, 21, 7, 46]. Nonetheless, these methods often perform suboptimally when applied to photo collections [32]. In contrast, our research aligns with efforts to synthesize static components from dynamic scenes.

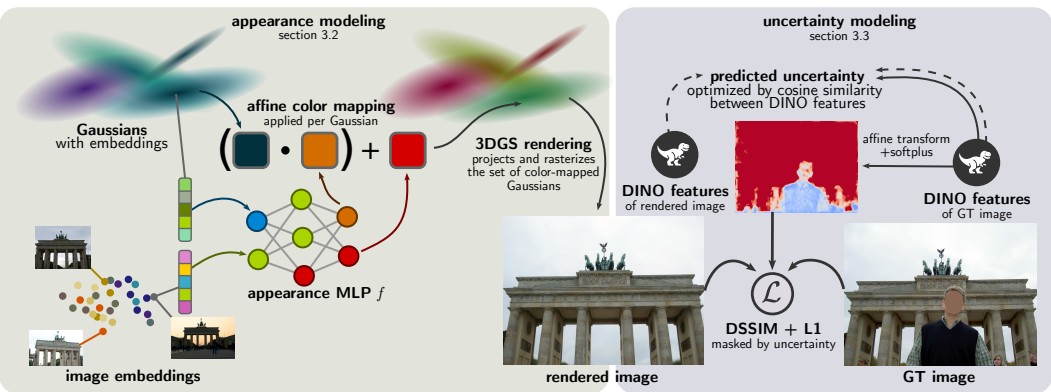

Figure 2: **Overview over the core components of WildGaussians.** *Left:* appearance modeling (Sec. 3.2). Per-Gaussian and per-image embeddings are passed as input to the appearance MLP which outputs the parameters of an affine transformation applied to the Gaussian's view-dependent color. *Right:* uncertainty modeling (Sec. 3.3). An uncertainty estimate is obtained by a learned transformation of the GT image's DINO features. To train the uncertainty, we use the DINO cosine similarity (dashed lines).

Methods such as RobustNeRF[32] utilize Iteratively Reweighted Least Squares for outlier verification in small, controlled settings, while NeRF *On-the-go* [31] employs DINO v2 features [27] to predict uncertainties, allowing it to handle complex scenes with varying occlusion levels, albeit with long training times. Unlike these approaches, our method optimizes significantly faster. Moreover, we effectively handle dynamic scenarios even with changes in illumination.

**Novel View Synthesis for Unstructured Photo Collections.** In real-world scenes, e.g. the unstructured internet photo collections [35], difficulties arise not only from dynamic occlusions like moving pedestrians and vehicles but also from varying illumination. Previously, these issues were tackled using multi-plane image (MPI) methods [20]. More recently, NeRF-W [24], a pioneering work in this area, addresses these challenges with per-image transient and appearance embeddings, along with leveraging aleatoric uncertainty for transient object removal. However, the method suffers from slow training and rendering speeds. Other NeRF-based methods followed NeRF-W extending it in various ways [37, 47]. Recent concurrent works, including our own, explore the replacement of NeRF representations with 3DGS for this task. Some methods [33, 6] address the simpler problem of training 3DGS under heavy occlusions, or only tackling appearance changes [23, 48, 8] with no occlusions. However, the main challenge is integrating appearance conditioning with the locally independent 3D Gaussians under occlusions. VastGaussian [22] applies a convolutional network to 3DGS outputs which does not transfer to large appearance changes, as shown in the Appendix. SWAG [5] and Scaffold-GS [23] address this by storing appearance data in an external hash-grid-based implicit field [26], while GS-W [52] and WE-GS [41] utilize CNN features for appearance conditioning on a reference image. In contrast, our method employs a simpler and more scalable strategy by embedding appearance vectors directly within each Gaussian. This design not only simplifies the architecture but also enables us to 'bake' the trained representation back into 3DGS after appearances are fixed, enhancing both efficiency and adaptability. Finally, a concurrent work, Splatfacto-W [45], uses a similar appearance MLP to combine Gaussian and image embeddings to output spherical harmonics.

## 3   Method

Our approach, termed WildGaussians, is shown in Fig. 2. To allow 3DGS-based approaches to handle the uncontrolled capture of scenes, we propose two key components: (1) **appearance modeling** enables our approach to handle the fact that the observed pixel colors not only depend on the viewpoint but also on conditions such as the capture time and the weather. Following NeRF-based approaches for reconstructing scenes from images captured under different conditions [24, 30], we train an appearance embedding per training image to model such conditions. In addition, we train an appearance embedding per Gaussian to model local effects, e.g., active illumination of parts of the scene from lamps. Both embeddings are used to transform the color stored for a Gaussian to match the color expected for a given scene appearance. To this end, we predict an affine mapping

[30] in color space via an MLP. (2) **uncertainty modeling** allows our approach to handle occluders during the training stage by determining which regions of a training image should be ignored. To this end, we extract DINO v2 features [27] from training images, and pass them as input to a trainable affine transformation which predicts a per-pixel uncertainty, encoding which parts of an image likely correspond to static regions and which parts show occluders. The uncertainty predictor is optimized using the cosine similarity between the DINO features extracted from training images and renderings.

## 3.1 Preliminaries: 3D Gaussian Splatting (3DGS)

We base our method on the 3D Gaussian Splatting (3DGS) [14, 50] scene representation, where the scene is represented as a set of 3D Gaussians $\{\mathcal{G}_i\}$. Each Gaussian $\mathcal{G}_i$ is represented by its mean $\mu_i$, a positive semi-definite covariance matrix $\Sigma_i$ [54], an opacity $\alpha_i$, and a view-dependent color parametrized using spherical harmonics (SH). During rendering, the 3D Gaussians are first projected into the 2D image [54], resulting in 2D Gaussians. Let $W$ be the viewing transformation, then the 2D covariance matrix $\Sigma'_i$ in image space is given as [54]:

$$\Sigma'_i = \left( JW\Sigma_i W^T J^T \right)_{1:2,1:2} \; , \tag{1}$$

where $J$ is the Jacobian of an affine approximation of the projection. $(\cdot)_{1:2,1:2}$ denotes the first two rows and columns of a matrix. The 2D Gaussian's mean $\mu'_i$ is obtained by projecting $\mu_i$ into the image using $W$. After projecting the Gaussians, the next step is to compute a color value for each pixel. For each pixel, the list of Gaussians is traversed from front to back (ordered based on the distances of the Gaussians to the image plane), alpha-compositing their view-dependent colors $\hat{c}_i(\mathbf{r})$ (where $\mathbf{r}$ is the ray direction corresponding to the pixel), resulting in pixel color $\hat{C}$:

$$\hat{C} = \sum_i \alpha_i \hat{c}_i(\mathbf{r}) \,, \qquad \text{with} \qquad \alpha_i = e^{\frac{1}{2}(x-\mu'_i)^T (\Sigma'_i)^{-1} (x-\mu'_i)} \; , \tag{2}$$

where $\alpha_i$ are the blending weights. The representation is learned from a set of images with known projection matrices using a combination of DSSIM [42] and L1 losses computed between the predicted colors $\hat{C}$ and ground truth colors $C$ (as defined by the pixels in the training images):

$$\mathcal{L}_{\text{3DGS}} = \lambda_{\text{dssim}} \text{DSSIM}(\hat{C}, C) + (1 - \lambda_{\text{dssim}}) \|\hat{C} - C\|_1 \,. \tag{3}$$

3DGS [14] further defines a process in which unused Gaussians with a low $\alpha_i$ or a large 3D size are pruned and new Gaussians are added by cloning or splitting Gaussians with large gradient wrt. 2D means $\mu'_i$. In our work, we further incorporate two recent improvements. First, the 2D $\mu'_i$ gradients are accumulated by accumulating absolute values of the gradients instead of actual gradients [49, 51]. Second, we use Mip-Splatting [50] to reduce aliasing artifacts.

## 3.2 Appearance Modeling

Following the literature on NeRFs [24, 30, 1], we use trainable per-image embeddings $\{\mathbf{e}_j\}_{j=1}^{N}$, where $N$ is the number of training images, to handle images with varying appearances and illuminations, such as those shown in Fig. 1. Additionally, to enable varying colors of Gaussians under different appearances, we include a trainable embedding $\mathbf{g}_i$ for each Gaussian $i$. We input the per-image embedding $\mathbf{e}_j$, per-Gaussian embedding $\mathbf{g}_i$, and the base color $\bar{c}_i$ (0-th order SH) into an MLP $f$:

$$(\beta, \gamma) = f(\mathbf{e}_j, \mathbf{g}_i, \bar{c}_i) \; . \tag{4}$$

The output are the parameters of an affine transformation, where $(\beta, \gamma) = \{(\beta_k, \gamma_k)\}_{k=1}^{3}$ for each color channel $k$. Let $\hat{c}_i(\mathbf{r})$ be the $i$-th Gaussian's view-dependent color conditioned on the ray direction $\mathbf{r}$. The toned color of the Gaussian $\tilde{c}_i$ is given as:

$$\tilde{c}_i = \gamma \cdot \hat{c}_i(\mathbf{r}) + \beta \; . \tag{5}$$

These per-Gaussian colors then serve as input to the 3DGS rasterization process. Our approach is inspired by [30], which predicts the affine parameters from the image embedding alone in order to compensate for exposure changes in images. In contrast, we use an affine transformation to model much more complex changes in appearance. In this setting, we found it necessary to also use per-Gaussian appearance embeddings to model local changes such as parts of the scene being actively illuminated by light sources at night.

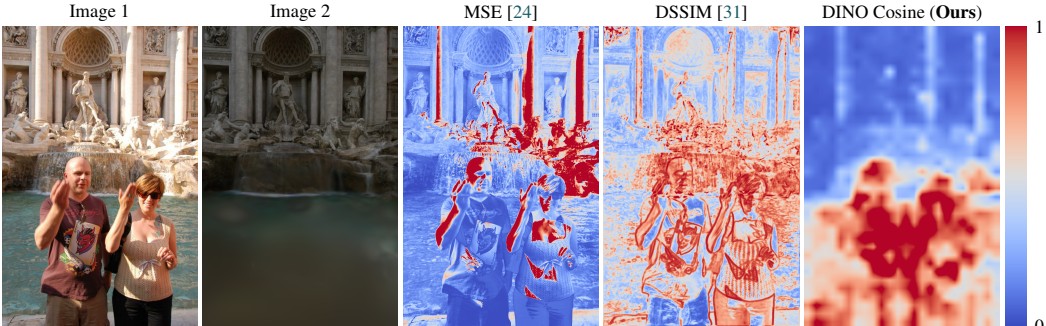

| Image 1 | Image 2 | MSE [24] | DSSIM [31] | DINO Cosine (**Ours**) |

Figure 3: **Uncertainty Losses Under Appearance Changes.** We compare MSE and DSSIM uncertainty losses (used by NeRF-W [24] and NeRF *On-the-go* [31]) to our DINO cosine similarity loss. Under heavy appearance changes (as in Image 1 and 2), both MSE and DSSIM fail to focus on the occluder (humans) and falsely downweight the background, while partly ignoring the occluders.

Note that if rendering speed is important at test time and the scene only needs to be rendered under a single static condition, it is possible to pre-compute the affine parameters per Gaussian and use them to update the Gaussian's SH parameters. This essentially results in a standard 3DGS representation [14, 50] that can be rendered efficiently.

**Initialization of Per-Gaussian Embeddings $\mathbf{g}_i$.** Initializing the embedding $\mathbf{g}_i$ randomly could lead to a lack of locality bias, and thus poorer generalization and training performance, as shown in the *supp. mat.* Instead, we initialize them using Fourier features [25, 40] to enforce a locality prior: We first center and normalize the input point cloud to the range of $[0, 1]$ using the $0.97$ quantile of the $L^\infty$ norm. The Fourier features of a normalized point $p$ are then obtained as a concatenation of $\sin(\pi p_k 2^m)$ and $\cos(\pi p_k 2^m)$, where $k = 1, 2, 3$ are the coordinate indices and $m = 1, \ldots, 4$.

**Training Objective.** Following 3DGS [14], we use a combination of DSSIM [42] and L1 losses for training (Eq. (3)). However, DSSIM and L1 are used for different purposes in our case. For DSSIM, since it is more robust than L1 to appearance changes and focuses more on structure and perceptual similarity, we apply it to the image rasterized without appearance modeling. On the other hand, we use the L1 loss to learn the correct appearance. Specifically, let $\hat{C}$ and $\tilde{C}$ be the rendered colors of the rasterized image before and after the color toning (cf. Eq. (5)), respectively. Let $C$ be the training RGB image. The training loss can be written as:

$$\mathcal{L}_{\text{color}} = \lambda_{\text{dssim}} \text{DSSIM}(\hat{C}, C) + (1 - \lambda_{\text{dssim}}) \|\tilde{C} - C\|_1 \ . \tag{6}$$

In all our experiments we set $\lambda_{\text{dssim}} = 0.2$. During training, we first project the Gaussians into the 2D image plane, compute the toned colors, and then rasterize the two images (toned and original colors).

**Test-Time Optimization of Per-Image Embeddings $\mathbf{e}_j$.** During training, we jointly optimize the per-image embeddings $\mathbf{e}_j$ and the per-Gaussian embedding $\mathbf{g}_i$, jointly with the 3DGS representation and the appearance MLP. However, when we want to fit the appearance of a previously unseen image, we need to perform test-time optimization of the unseen image's embedding. To do so, we initialize the image's appearance vector as zeroes and optimize it with the main training objective (Eq. (6)) using the Adam optimizer [16] - while keeping everything else fixed.

### 3.3 Uncertainty Modeling for Dynamic Masking

To reduce the influence of transient objects and occluders, e.g., moving cars or pedestrians, on the training process we learn an uncertainty model [24, 31]. NeRF *On-the-go* [31] showed that using features from a pre-trained feature extractor, e.g., DINO [3, 27], increases the robustness of the uncertainty predictor. However, while working well in controlled settings, the uncertainty loss function cannot handle strong appearance changes (such as these in unconstrained image collections). Therefore, we propose an alternative uncertainty loss which is more robust to appearance changes as can be seen in Figure 3. During training, for each training image $j$, we first extract DINO v2 [27] features. Then, our uncertainty predictor is simply a trainable affine mapping applied to the DINO features, followed by the softplus activation function. Since the features are patch-wise ($14 \times 14$px),

we upscale the resulting uncertainty to the original size using bilinear interpolation. Finally, we clip the uncertainty to the interval $[0.1, \infty)$ to ensure a minimal weight is assigned to each pixel [24, 31].

**Uncertainty Optimization.** In the NeRF literature [24, 31], uncertainty modeling is realized by letting the model output a Gaussian distribution for each pixel instead of a single color value. For each pixel, let $\tilde{C}$ and $C$ be the predicted and ground truth colors. Let $\sigma$ be the (predicted) uncertainty. The per-pixel loss function is the (shifted) negative log-likelihood of the normal distribution with mean $\tilde{C}$, and variance $\sigma$ [24, 31]:

$$\mathcal{L}_u = -\log\left(\frac{1}{\sqrt{2\pi\sigma^2}}\exp\left(-\frac{\|\tilde{C}-C\|_2^2}{2\sigma^2}\right)\right) = \frac{\|\tilde{C}-C\|_2^2}{2\sigma^2} + \log\sigma + \frac{\log 2\pi}{2} \ . \tag{7}$$

In [31], the squared differences are replaced by a slightly modified DSSIM, which was shown to have benefits over the MSE loss used in [24]. Even though DSSIM has a different distribution than MSE [42, 2], [31] showed that it can lead to stable training dynamics. Unfortunately, as shown in Fig. 3, both MSE and DSSIM are not robust to appearance changes. This prevents these MSE-based and SSIM-based methods [24, 31] from learning the correct appearance as the regions with varying appearances are ignored by the optimization process. However, we can once more take advantage of DINO features which are more robust to appearance changes, and construct our loss function from the cosine similarity between the DINO features of the training image and the predicted image. Since DINO features are defined per image patch, not pixel, we compute our uncertainty loss per patch. Let $\tilde{D}$ and $D$ be the DINO features of the predicted and the training image patch. The loss is as follows:

$$\mathcal{L}_{\text{dino}}(\tilde{D}, D) = \min\left(1, 2 - \frac{2\tilde{D}\cdot D}{\|\tilde{D}\|_2\|D\|_2}\right) \ , \tag{8}$$

where '·' denotes the dot product. Note, that this loss function will be zero when the two features have a cosine similarity of 1, and it will become 0 when the similarity drops below $1/2$.

Finally, to optimize uncertainty, we also add the log prior resulting in the following per-patch loss:

$$\mathcal{L}_{\text{uncertainty}} = \frac{\mathcal{L}_{\text{dino}}(\tilde{D}, D)}{2\sigma^2} + \lambda_{\text{prior}}\log\sigma \ , \tag{9}$$

where $\sigma$ is the uncertainty prediction for the patch. We use this loss only to optimize the uncertainty predictor (implemented as a single affine transformation) without letting the gradients propagate through the rendering pipeline [31]. Further, during 3DGS training, the opacity is periodically reset to a small value to prevent local minima. However, after each opacity reset, the renderings are corrupted by (temporarily) incorrect alphas. To prevent this issue from propagating into the uncertainty predictor, we hence disable the uncertainty training for a few iterations after each opacity reset.

**Optimizing 3DGS with Uncertainty.** For NeRFs, one can use the uncertainty to directly weight the training objective [24, 31]. In our experiments, we observed that this would not lead to stable training as the absolute values of gradients are used in the densification algorithm and the excessively large magnitudes of absolute gradients lead to excessive growth. The uncertainty weighting would make the setup sensitive to the correct choice of hyperparameters. Therefore, to handle this issue, we propose to convert the uncertainty scores into a (per-pixel) binary mask such that the gradient scaling is at most one:

$$M = \mathbb{1}\left(\frac{1}{2\sigma^2} > 1\right) \ , \tag{10}$$

where $\mathbb{1}$ is the indicator function which is 1 whenever the uncertainty multiplier is greater than 1. This mask is then used to multiply the per-pixel loss defined in Eq. (6):

$$\mathcal{L}_{\text{color-masked}} = \lambda_{\text{dssim}}M\,\text{DSSIM}(\hat{C}, C) + (1 - \lambda_{\text{dssim}})M\|\tilde{C} - C\|_1 \ . \tag{11}$$

### 3.4 Handling Sky

For realistic renderings of a scene under different conditions, modeling the sky is important (see Fig. 1). It is unlikely that Gaussians are created in the sky when using Structure-from-Motion points as initialization. Thus, we sample points on a sphere around the 3D scene and add them to the set of

| Method | GPU hrs. /FPS | Low Occlusion | | | Medium Occlusion | | | High Occlusion | | | Average | | |
|---|---|---|---|---|---|---|---|---|---|---|---|---|---|
| | | PSNR↑ | SSIM↑ | LPIPS↓ | PSNR↑ | SSIM↑ | LPIPS↓ | PSNR↑ | SSIM↑ | LPIPS↓ | PSNR↑ | SSIM↑ | LPIPS↓ |
| NeRF *On-the-go* [31] | 43/<1 | 20.63 | 0.661 | 0.191 | 22.31 | 0.780 | 0.130 | 22.19 | 0.753 | 0.169 | 21.71 | 0.731 | 0.163 |
| 3DGS [14] | 0.35/116 | 19.68 | 0.649 | 0.199 | 19.19 | 0.709 | 0.220 | 19.03 | 0.649 | 0.340 | 19.30 | 0.669 | 0.253 |
| Gaussian Opacity Fields [51] | *0.41/43** | 20.54 | 0.662 | 0.178 | 19.39 | 0.719 | 0.231 | 17.81 | 0.578 | 0.430 | 19.24 | 0.656 | 0.280 |
| Mip-Splatting [50] | *0.18/82** | 20.15 | 0.661 | 0.194 | 19.12 | 0.719 | 0.221 | 18.10 | 0.664 | 0.333 | 19.12 | 0.681 | 0.249 |
| GS-W [52] | *0.55/71** | 18.67 | 0.595 | 0.288 | 21.50 | 0.783 | 0.152 | 18.52 | 0.644 | 0.335 | 19.56 | 0.674 | 0.258 |
| **Ours** | 0.50/108 | 20.62 | 0.658 | 0.235 | 22.80 | 0.811 | 0.092 | 23.03 | 0.771 | 0.172 | 22.15 | 0.756 | 0.167 |

\* Methods were trained and evaluated on NVIDIA A100, while the rest used NVIDIA GTX 4090.

Table 1: **Comparison on NeRF *On-the-go* Dataset [31].** The first , second , and third values are highlighted. Our method shows overall superior performance over state-of-the-art baseline methods.

| | GPU hrs./ FPS | Brandenburg Gate | | | Sacre Coeur | | | Trevi Fountain | | |
|---|---|---|---|---|---|---|---|---|---|---|
| | | PSNR ↑ | SSIM ↑ | LPIPS ↓ | PSNR ↑ | SSIM ↑ | LPIPS ↓ | PSNR ↑ | SSIM ↑ | LPIPS ↓ |
| NeRF [25] | -/<1 | 18.90 | 0.815 | 0.231 | 15.60 | 0.715 | 0.291 | 16.14 | 0.600 | 0.366 |
| NeRF-W-re [24] | 164/<1 | 24.17 | 0.890 | 0.167 | 19.20 | 0.807 | 0.191 | 18.97 | 0.698 | 0.265 |
| Ha-NeRF [4] | 452/<1 | 24.04 | 0.877 | 0.139 | 20.02 | 0.801 | 0.171 | 20.18 | 0.690 | 0.222 |
| K-Planes [9] | 0.6/<1 | 25.49 | 0.879 | 0.224 | 20.61 | 0.774 | 0.265 | 22.67 | 0.714 | 0.317 |
| RefinedFields [13] | 150/<1 | 26.64 | 0.886 | - | 22.26 | 0.817 | - | 23.42 | 0.737 | - |
| 3DGS [14] | 2.2/57 | 19.37 | 0.880 | 0.141 | 17.44 | 0.835 | 0.204 | 17.58 | 0.709 | 0.266 |
| GS-W[†] [52] | 1.2/51 | 23.51 | 0.897 | 0.166 | 19.39 | 0.825 | 0.211 | 20.06 | 0.723 | 0.274 |
| SWAG[*] [5] | 0.8/15 | 26.33 | 0.929 | 0.139 | 21.16 | 0.860 | 0.185 | 23.10 | 0.815 | 0.208 |
| **Ours** | 7.2/117 | 27.77 | 0.927 | 0.133 | 22.56 | 0.859 | 0.177 | 23.63 | 0.766 | 0.228 |

[†] Evaluated using NeRF-W evaluation protocol [24, 18]; [*] Source code not available, numbers from the paper, unknown GPU used.

Table 2: **Comparison on the Photo Tourism Dataset [35].** The first , second , and third best-performing methods are highlighted. We significantly outperform all baseline methods and offer the fastest rendering times.

points that is used to initialize the 3D Gaussians. For an even distribution of points on the sphere, we utilize the Fibonacci sphere sampling algorithm [36], which arranges points in a spiral pattern using a golden ratio-based formula. After placing these points on the sphere at a fixed radius $r_s$, we project them to all training cameras, removing any points not visible from at least one camera. Details are included in the *supp. mat.*

## 4 Experiments

**Datasets.** We evaluate our WildGaussians approach on two challenging datasets. The **NeRF *On-the-go* dataset [31]** contains multiple casually captured indoor and outdoor sequences, with varying ratios of occlusions (from 5% to 30%). For evaluation, the dataset provides 6 sequences in total. Note that there are almost no illumination changes across views in this dataset. Since 3DGS [14] cannot handle radially distorted images, we train and evaluate our method and all baselines on a version of the dataset where all images were undistorted. The **Photo Tourism dataset [35]** consists of multiple 3D scenes of well-known monuments. Each scene has an unconstrained collection of images uploaded by users captured at different dates and times of day with different cameras and exposure levels. In our experiments we use the Brandenburg Gate, Sacre Coeur, and Trevi Fountain scenes, which have an average ratio of occlusions of 3.5%. Note, that for each dataset (both NeRF *On-the-go* and Photo Tourism), the test set was carefully chosen not to have any occluders.

**Baselines.** We compare our approach against a set of baselines. We use NerfBaselines [18] as our evaluation framework, providing a unified interface to the original released source codes while ensuring fair evaluation. On the NeRF *On-the-go* dataset, which contains little illumination changes, we compare to NeRF *On-the-go* [31], the original 3DGS formulation [14], Mip-Splatting [50], and Gaussian Opacity Fields [51]. On the Photo Tourism dataset [35], we compare against the most recent state-of-the-art methods for handling strong illumination changes: NeRF-W-re [24] (open source implementation), Ha-NeRF [4], K-Planes [9], RefinedFields [13], 3DGS [14], and concurrent works

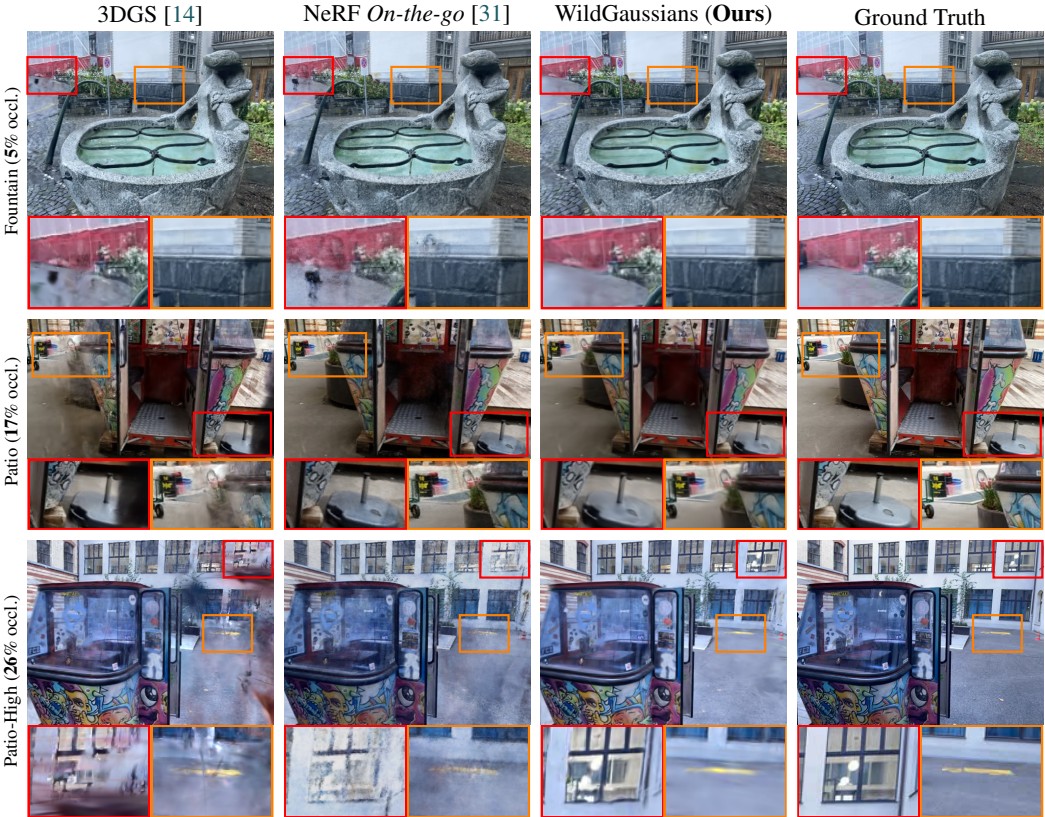

| 3DGS [14] | NeRF *On-the-go* [31] | WildGaussians (**Ours**) | Ground Truth |

Figure 4: **Comparison on NeRF *On-the-go* Dataset [31]**. For both the Fountain and Patio-High scenes, we can see that the baseline methods exhibit different levels of artifacts in the rendering, while our method removes all occluders and shows the best view synthesis results.

GS-W [52] and SWAG [5]. We evaluate GS-W [52] using the NeRF-W evaluation protocol [24] (see below). Our GS-W numbers thus differ from those in [52] (which conditioned on full test images).

**Metrics.** We follow common practice and use PSNR, SSIM [42], and LPIPS [53] for our evaluation. For the Photo Tourism dataset [35], we use the evaluation protocol proposed in NeRF-W [24, 18], where the image appearance embedding is optimized on the left half of the image. The metrics are then computed on the right half. For the NeRF *On-the-go* dataset [31], there is no test-time optimization. We also report training times in GPU hours as well as rendering times in frames-per-second (FPS), computed on an NVIDIA RTX 4090 unless stated otherwise.

### 4.1   Comparison on the NeRF *On-the-go* Dataset

As shown in Table 1 and Fig. 4, our approach significantly outperforms both baselines, especially for scenarios with medium (15-20%) to high occlusions (30%). Compared to NeRF *On-the-go* [31], our method is not only $400\times$ faster in rendering, but can more effectively remove occluders. Moreover, we can better represent distant and less-frequently seen background regions (first and third row in Fig. 4). Interestingly, 3DGS and its derivatives (Mip-Splatting, Gaussian Opacity Fields) are quite robust to scenes with low occlusion ratios, thanks to its geometry prior in the form of the initial point cloud. Nevertheless, 3DGS and its derivatives struggles to remove occlusions for high-occlusion scenes. This demonstrates the effectiveness of our uncertainty modeling strategy.

### 4.2   Comparision on Photo Tourism

Table 2 and Fig. 5 show results on the challenging Photo Tourism dataset. As for the NeRF *On-the-go* dataset, our method shows notable improvements over all NeRF-based baselines, while enabling real-time rendering (similar to 3DGS). Compared to 3DGS, we can adeptly handle changes in appearance

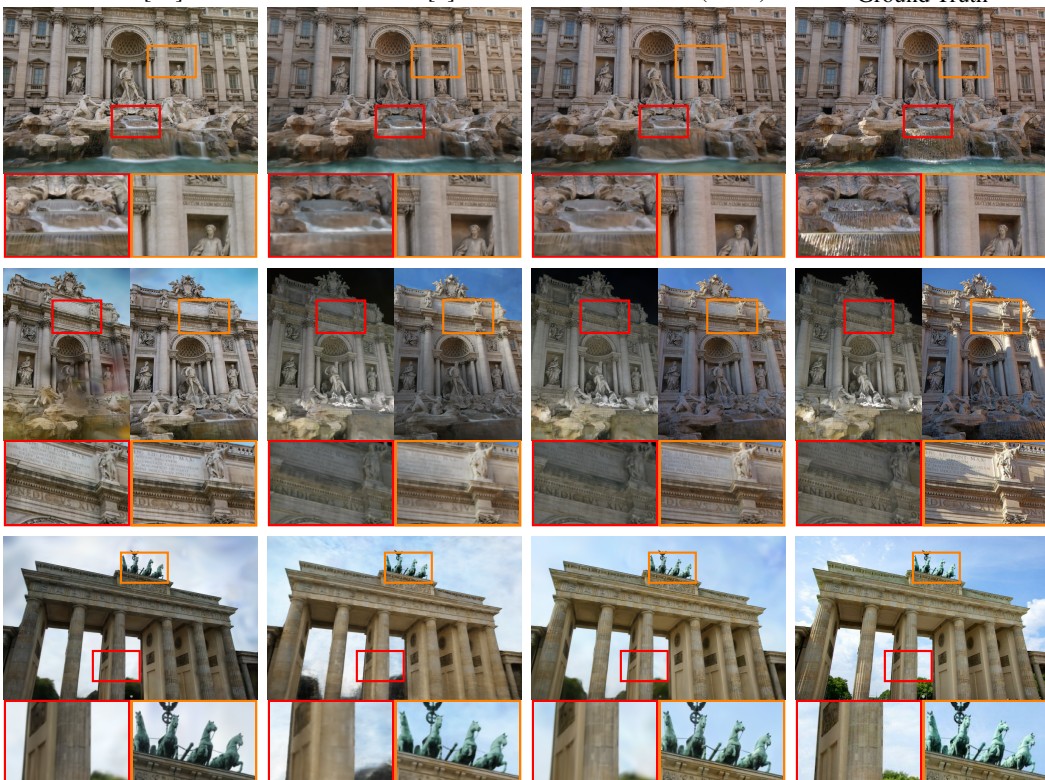

| 3DGS [14] | K-Planes [9] | WildGaussians (**Ours**) | Ground Truth |

Figure 5: **Comparison on the Photo Tourism Dataset [35].** In the first row, note that while none of the methods can represent the reflections and details of the flowing water, 3DGS and WildGaussians can provide at least some details even though there are no multiview constraints on the flowing water. On the second row, notice how 3DGS tries to 'simulate' darkness by placing dark - semi-transparent Gaussians in front of the cameras. For WildGaussians, the text on the building is legible. WildGaussians is able to recover fine details in the last row.

| Dataset (**ratio**) | MipNeRF360 [1] (**0%**) | | | Photo Tourism [35] (**3.5%**) | | | *On-the-go* [31] low. (**5%**) | | | *On-the-go* [31] high. (**26%**) | | |
|---|---|---|---|---|---|---|---|---|---|---|---|---|
| | PSNR↑ | SSIM↑ | LPIPS↓ | PSNR↑ | SSIM↑ | LPIPS↓ | PSNR↑ | SSIM↑ | LPIPS↓ | PSNR↑ | SSIM↑ | LPIPS↓ |
| **Ours** | 23.73 | 0.643 | 0.304 | 24.63 | 0.851 | 0.179 | 20.62 | 0.658 | 0.235 | 23.03 | 0.771 | 0.172 |
| w/o uncert. | 23.71 | 0.643 | 0.313 | 24.32 | 0.845 | 0.187 | 20.53 | 0.646 | 0.229 | 20.27 | 0.652 | 0.337 |
| w/o app. | 23.31 | 0.641 | 0.319 | 18.47 | 0.794 | 0.241 | 20.80 | 0.669 | 0.228 | 22.80 | 0.755 | 0.183 |

Table 3: We conduct **ablation studies** on the Photo Tourism [35], NeRF *On-the-go* [31], and MipNeRF360 (bicycle) [1] datasets with varying degree of occlusion. The first , second , and third values are highlighted.

such as day-to-night transitions without sacrificing fine details. This shows the efficacy of our appearance modeling. Compared to the NeRF-based baseline K-Planes [9], our method offers shaper details, as can be noticed in the flowing water and text on the Trevi Fountain. Compared to 3DGS [14], our method has comparable rendering speed on the NeRF *On-the-go* dataset, while being much faster on the Photo Tourism dataset [35]. This is caused by 3DGS [14] trying to grow unnecessary Gaussians to explain higher gradients due to the appearance variation. Finally, compared to other 3DGS-based methods [52, 5], ours achieves stronger performance while having faster inference because we can 'bake' the apperance-tuned spherical harmonics back into the standard 3DGS representation.

## 4.3 Ablation Studies & Analysis

To validate the importance of each component of our method, we conduct an ablation study in Table 3, separately disabling either uncertainty or appearance modeling. Table 3 shows that without **appearance modeling**, performance significantly drops on the Photo Tourism dataset due to the strong appearance changes captured by the dataset. On the NeRF *On-the-go* dataset, which exhibit

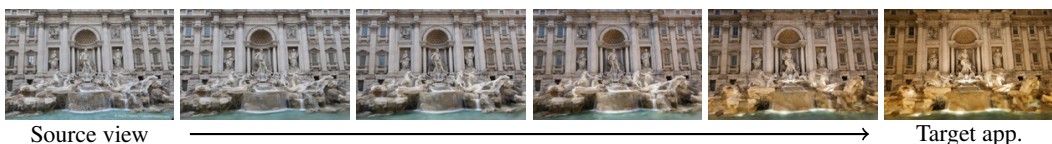

Source view ⟶ Target app.

Figure 6: **Appearance interpolation.** We show how the appearance changes as we interpolate from a (*daytime*) view to a (*nighttime*) view's appearance. Notice the light sources gradually appearing.

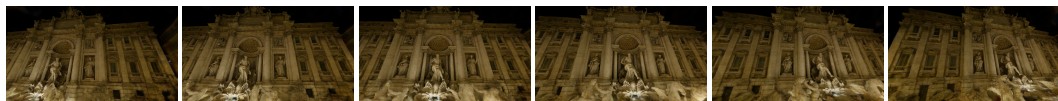

Figure 7: **Fixed appearance multi-view consistency**. We shows the multiview consistency of a fixed *nighttime* appearance embedding as the camera moves around the fountain.

little to no illumination or other appearance changes, disabling appearance modeling only slightly improves performance. We conclude that it is safe to use appearance embeddings, even if there might not be strong appearance changes. Similarly, disabling uncertainty modeling has little impact on datasets with less occlusions and could even make the performance slightly worse for the *On-the-go* low dataset, but it is required for the high-occlusion datasets (*On-the-go* high and Photo Tourism).

As expected, for dataset with low occlusion ratios, disabling **uncertainty modeling** has a limited impact on the overall performance. We attribute this to the inherent robustness of 3DGS, where the initial 3D point clouds also help to filter out some occlusions. However, as the occlusion ratio increases, the importance of uncertainty modeling becomes evident. This is shown by the significant performance drop when using no uncertainty modeling for the NeRF *On-the-go* high occlusion dataset.

**Behavior of the appearance embedding.** Fig. 6 interpolates between two appearance embeddings. The transition from the source view to the target view's appearance is smooth, with lights gradually appearing. This demonstrates the smoothness and the continuous nature of the embedding space. In Fig. 7, we interpolate between two camera poses with a fixed appearance embedding showing multiview consistency. Next, we further analyze the embedding space with a t-SNE [39] projection of the embeddings of

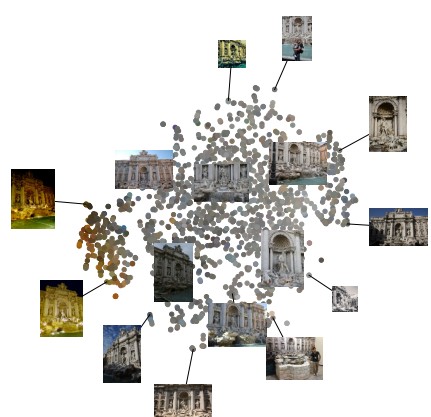

Figure 8: **t-SNE for Appearance Embedding.** We visualize the training images' appearance embeddings using t-SNE. See the day/night separation.

training images. The t-SNE visualization in Fig. 8 reveals that the embeddings are grouped by image appearances, e.g., with night images clustering together and being separated from other images.

## 5 Conclusion

Our WildGaussians model extends Gaussian Splatting to uncontrolled in-the-wild settings where images are captured across different time or seasons, normally with occluders of different ratios. The key to the success are our novel appearance and uncertainty modeling tailored for 3DGS, which also ensures high-quality real-time rendering. We believe our method is a step toward achieving robust and versatile photorealistic reconstruction from noisy, crowd-sourced data sources.

**Limitations.** While our method enables appearance modeling with real-time rendering, it is currently not able to capture highlights on objects. Additionally, although the uncertainty modeling is more robust than MSE or SSIM, it still struggles with some challenging scenarios. If there are not enough observations of a part of the scene, e.g., because it is occluded in nearly all training images, our approach will struggle to correctly reconstruct the region. One way to handle this is to incorporate additional priors such as pre-trained diffusion models. We leave it as future work.

## Acknowledgments and Disclosure of Funding

We would like to thank Weining Ren for his help with the NeRF On-the-go dataset and code and Tobias Fischer and Xi Wang for fruitful discussions. This work was supported by the Czech Science Foundation (GAČR) EXPRO (grant no. 23-07973X), and by the Ministry of Education, Youth and Sports of the Czech Republic through the e-INFRA CZ (ID:90254). Jonas Kulhanek acknowledges travel support from the European Union's Horizon 2020 research and innovation programme under ELISE Grant Agreement No 951847.

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

# A    Appendix / Supplemental Material

## A.1    Implementation & Experimental Details

We base our implementation on the INRIA's original 3DGS renderer [14], extended in Mip-Splatting [50]. Furthermore, we extended the implementation by the AbsGaussian/Gaussian Opacity Fields absolute gradient scaling fix [51, 49]. All our experiments were conducted on a single NVIDIA RTX 4090 GPU. For the uncertainty modeling, we use the ViT-S/14 DINO v2 features (smallest DINO configuration) [27]. We also resize the images before computing DINO features to the max. size of 350 to make the DINO loss computation faster. To model the appearance, we use embeddings of size 24 for the Gaussians and 32 for the image appearance embeddings. For the appearance MLP, we use 2 hidden layers of size 128. We use the ReLU activation function. We use the Adam optimizer [16] without weight decay. For test-time appearance embedding optimization, we perform 128 Adam's gradient descent steps with the learning rate of 0.1. For the main training objective $\lambda_{dssim}$ to 0.2 and $\lambda_{uncert}$ to 0.5. We now describe the hyper-parameters used for the two datasets (Photo Tourism [35], NeRF *On-the-go* [31]):

**NeRF *On-the-go* Dataset.** We optimize each representation for $30k$ training steps. For our choice of learning rates, we mostly follow 3DGS [14]. We differ in the following learning rates: appearance MLP lr. of 0.0005, uncertainty lr. of 0.001. Gaussian embedding lr. of 0.005. image embedding lr. of 0.001. For the position learning rate, we exponentially decay from $1.6 \times 10^{-4}$ to $1.6 \times 10^{-6}$. Furthermore, we set the densification threshold at 0.0002 and density from 500-th iteration to $15\,000$-th iteration every 100 steps. Furthermore, we reset the opacity every $3\,000$ steps. We do not optimize the uncertainty predictor for 500 after each opacity reset, and we do not apply the masking for the first 2000 training steps.

**Photo Tourism.** We optimize each representation for $200k$ training steps. We use the following learning rates: scales lr. of 0.0005, rotation lr. of 0.001, appearance MLP lr. of 0.0005, uncertainty lr. of 0.001. Gaussian embedding lr. of 0.005. image embedding lr. of 0.001. For the position learning rate, we exponentially decay from $1.6 \times 10^{-5}$ to $1.6 \times 10^{-7}$. We set the densification threshold at 0.0002 and density from 4000-th iteration to $100\,000$-th iteration every 400 steps. Furthermore, we reset the opacity every $3\,000$ steps. We do not optimize the uncertainty predictor for 1500 after each opacity reset. We do not use the uncertainty predictor for masking for the first $35\,000$ steps of the training, and after that, we linearly increase its contribution for $5\,000$ steps.

**Appearance MLP Priors.** Considering that the appearance MLP shares weights across all Gaussians, we observed potential instability in training when randomly initializing the MLP. To mitigate this, we introduce a prior to the appearance MLP $f_\theta$, optimizing gradient flow during early training phases. The raw output of the last layer of the MLP, $(\hat{\beta}, \hat{\gamma})$, is adjusted to obtain the affine color transformation in Eq. (5) as $\beta_k = 0.01\hat{\beta}_k$ and $\gamma_k = 0.01\hat{\gamma}_k + 1$. This adjustment scales both the initialization of the last layer and the learning rates of the MLP by 0.01, stabilizing early-stage training such that the gradient wrt. SH coefficients is predominant.

**Sky Initialization.** We set the scene radius $r_s$ to the 97% quantile of the L2 norms of the centered input 3D points. We then initialize a sphere at a distance of $10r_s$ from the scene center [15]. For an even distribution of points on the sphere, we utilize the Fibonacci sphere sampling algorithm [36],

| | Brandenburg Gate | | | Sacre Coeur | | | Trevi Fountain | | |
|---|---|---|---|---|---|---|---|---|---|
| | PSNR ↑ | SSIM ↑ | LPIPS ↓ | PSNR ↑ | SSIM ↑ | LPIPS ↓ | PSNR ↑ | SSIM ↑ | LPIPS ↓ |
| **WildGaussians (Ours)** | 27.77 | 0.927 | 0.133 | 22.56 | 0.859 | 0.177 | 23.63 | 0.766 | 0.228 |
| w/o Fourier feat. | 27.33 | 0.924 | 0.141 | 22.32 | 0.853 | 0.188 | 23.78 | 0.766 | 0.229 |
| w/o appearance modeling | 20.81 | 0.874 | 0.187 | 17.19 | 0.811 | 0.237 | 17.42 | 0.697 | 0.299 |
| w/o appearance&uncertainty | 20.04 | 0.875 | 0.187 | 17.59 | 0.819 | 0.228 | 17.81 | 0.701 | 0.287 |
| VastGaussian [22] | 25.65 | 0.910 | 0.159 | 20.52 | 0.820 | 0.225 | 20.40 | 0.716 | 0.283 |
| w/o Gaussian embeddings [30] | 25.18 | 0.908 | 0.156 | 18.68 | 0.790 | 0.250 | 19.49 | 0.723 | 0.275 |
| w/o uncertainty modeling | 27.16 | 0.920 | 0.143 | 21.97 | 0.851 | 0.191 | 23.82 | 0.765 | 0.228 |
| MSSIM uncert. | 27.32 | 0.911 | 0.167 | 21.19 | 0.817 | 0.237 | 21.20 | 0.717 | 0.295 |
| w. explicit masks | 26.87 | 0.923 | 0.138 | 22.24 | 0.855 | 0.186 | 23.84 | 0.765 | 0.231 |

Table 4: **Detailed Ablation Study** conducted on the Photo Tourism dataset [35]. The first , second , and third values are highlighted.

which arranges points in a spiral pattern using a golden ratio-based formula. We sample $100\,000$ points on the sphere and then project them to all training cameras, removing any points not visible from at least one camera. This set of sky points is then added to our initial point set, with their opacity initialized at 1.0, while the opacity of the rest is set at 0.1.

## A.2 Extended Results on NeRF *On-the-go* Dataset

For reference, we extend the averaged results for the NeRF *On-the-go* dataset [31], by giving detailed results for individual scenes. The results are presented in Table 5.

| Method | GPU hrs./FPS | Low Occlusion Mountain PSNR SSIM LPIPS | Low Occlusion Fountain PSNR SSIM LPIPS | Medium Occlusion Corner PSNR SSIM LPIPS | Medium Occlusion Patio PSNR SSIM LPIPS | High Occlusion Spot PSNR SSIM LPIPS | High Occlusion Patio-High PSNR SSIM LPIPS |
|---|---|---|---|---|---|---|---|
| NeRF *On-the-go* [31] | 43/<1 | 20.46 **0.661** 0.186 | 20.79 0.661 0.195 | 23.74 0.806 0.127 | 20.88 0.754 **0.133** | 22.80 0.800 **0.132** | 21.57 0.706 **0.205** |
| 3DGS [14] | 0.35/116 | 19.40 0.638 0.213 | 19.96 0.659 **0.185** | 20.90 0.713 0.241 | 17.48 0.704 0.199 | 20.77 0.693 0.316 | 17.29 0.604 0.363 |
| Mip-Splatting | 0.18/82* | 19.86 0.649 0.200 | 20.19 0.672 0.189 | 21.15 0.728 0.230 | 18.31 0.639 0.328 | 20.18 0.689 0.338 | 18.31 0.639 0.328 |
| Gaussian Opacity Fields | 0.41/43* | **20.70** **0.661** 0.169 | 20.37 0.662 0.187 | 21.53 0.739 0.241 | 15.58 0.491 0.536 | 20.03 0.683 0.324 | 15.58 0.491 0.536 |
| GS-W | 0.55/71* | 19.43 0.596 0.299 | 20.06 **0.723** 0.274 | 22.17 0.793 0.155 | 19.90 0.681 0.260 | 17.13 0.608 0.409 | 19.90 0.681 0.260 |
| **Ours** | 0.50/108 | 20.43 0.653 0.255 | **20.81** 0.662 0.215 | **24.16** **0.822** **0.045** | **21.44** **0.800** 0.138 | **23.82** **0.816** 0.138 | **22.23** **0.725** 0.206 |

\* Methods were trained and evaluated on NVIDIA A100, while the rest used NVIDIA GTX 4090.

Table 5: **Extended Results on the NeRF *On-the-go* Dataset.**

## A.3 Extended Ablation Study

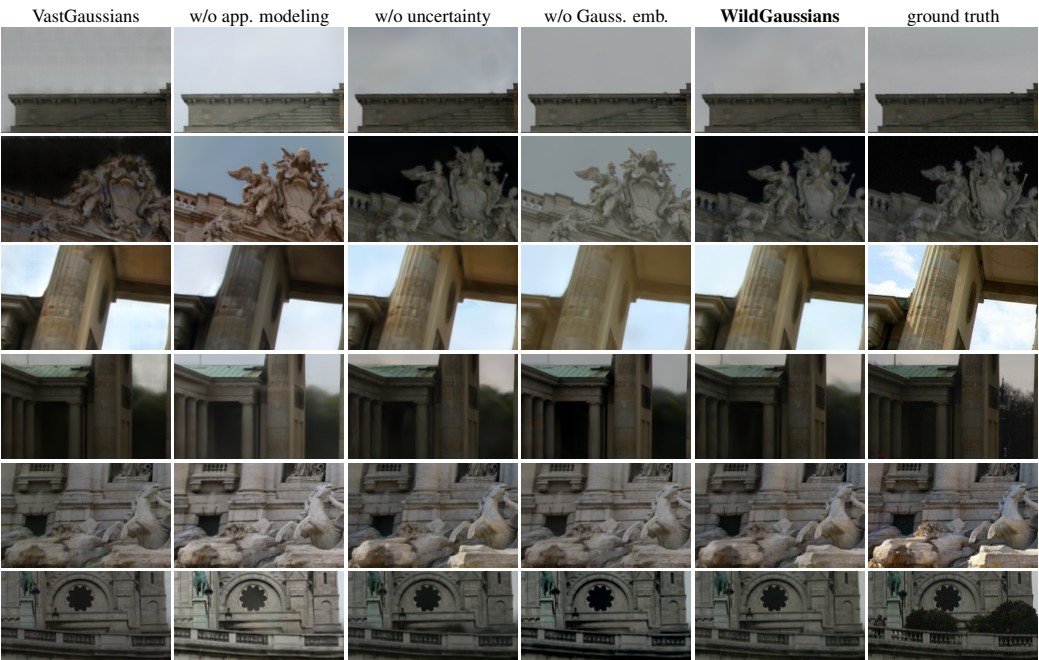

Figure 9: **Photo Tourism ablation study**. We show VastGaussian-style appearance modeling, no appearance modeling, no uncertainty modeling, no Gaussian embeddings (only per-image embeddings), and the full method.

To further analyze the performance of the proposed contributions, we performed a detailed ablation study of both the uncertainty prediction and appearance modeling. The results are presented in Table 4. As we can see from the table, WildGaussians appearance modeling outperforms other baselines. While VastGaussian [22] seems to work well for cases where the appearance difference is small, it fails in cases with large appearance changes and causes noticeable artifacts in the images. In the case of a simple affine color transformation (w/o Gaussian embeddings), this modeling is not

powerful enough to capture local appearance changes such as lamps turning on. We can also see the effectiveness of the uncertainty modeling. However, for the Trevi Fountain, the uncertainty does not improve the performance, which is likely caused by **1)** the scene has a small number of occlusions, **2)** the water in the scene is often mistaken for a transient object as it is not multi-view consistent. For reference, we also included a comparison with a method trained with explicit segmentation masks obtained from the MaskRCNN predictor [11].

We also present qualitative results in Figure 9. Notice how VastGaussians leave noticeable artifacts in the sky region in the first row. In the second row, notice how no appearance modeling or only using image embeddings cannot represent the dark sky. Similarly, not using Gaussian embeddings causes the method not to be able to represent shadows (row 2) and highlights (row 5). Disabling uncertainty leads to noticeable artifacts in row 6.

## A.4 Dataset occlusions

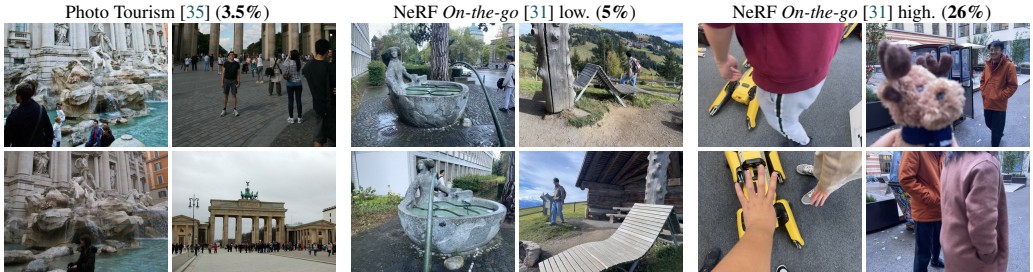

Figure 10: **Occluders** present in the Photo Tourism [35] and NeRF *On-the-go* [31] datasets.

To get some understanding of the types of occlusions present in the datasets, we visualize some images from the datasets with various amounts of occlusions in Figure 10. While for Photo Tourism [35], the occluders are humans looking into the camera in the bottom part of the images, for NeRF *On-the-go* [31], we have humans and objects present in various regions of the images.

## A.5 Licenses

Our renderer code is based on the 3DGS codebase [14] with the modifications from Mip-Splatting [50] and Gaussian Opacity Fields [51]. They are released under "research only" licenses: https://raw.githubusercontent.com/graphdeco-inria/gaussian-splatting/refs/heads/main/LICENSE.md, https://raw.githubusercontent.com/autonomousvision/mip-splatting/refs/heads/main/LICENSE.md, and https://raw.githubusercontent.com/autonomousvision/gaussian-opacity-fields/refs/heads/main/LICENSE.md.
The NeRF *On-the-go* dataset is licensed under the Apache 2.0 license (https://raw.githubusercontent.com/cvg/nerf-on-the-go/refs/heads/master/LICENSE). The pictures in the Photo Tourism dataset were sourced from various creators who made the images available under permissive licenses (https://creativecommons.org/share-your-work/cclicenses/). For details and the full list of creators, please refer to the original Photo Tourism paper [35].

