# OpenReview forum: "WildGaussians: 3D Gaussian Splatting In the Wild"
_NeurIPS.cc/2024/Conference — NeurIPS 2024 poster_

### Official Review · Reviewer_BAXf · 2024-07-06

**Soundness:** 3
**Presentation:** 3
**Contribution:** 2
**Rating:** 6
**Confidence:** 5

**Summary:**

This paper achieves in-the-wild reconstruction of 3D Gaussian splatting by introducing appearance embedding and DINO uncertainty mask. The appearance embedding is divided into a per-photo global embedding and a per-Gaussian local embedding. The uncertainty mask is obtained by comparing the rendered image with the ground truth DINO feature maps.

**Strengths:**

1. The presentation of this paper is very clear and easy to understand.
2. The performance in removing occluders is impressive.

**Weaknesses:**

1. This paper lacks innovation; both appearance embedding and the uncertainty mask are derived from previous work. Appearance embedding has already been demonstrated in SWAG, and using DINO features to address uncertainty is explicitly mentioned as coming from NeRF On-the-go.

2. In the section "Test-Time Optimization of Per-Image Embeddings," the authors mention using test images to optimize per-image embedding. In the original NeRF in-the-wild paper, their experimental setting involves using the left half of an image to optimize the embedding and evaluating it on the right half. Clearly, there is an unfair comparison in your experiments. Based on my experience, using the full image for optimization during the test, even if the rest of the 3DGS remains unchanged, can lead to overfitting the per-image embedding to the corresponding test image. It would be beneficial to include experimental data using the half-image optimization approach and conduct a fair comparison.

3. The qualitative results lack the demonstration of applying a single per-image embedding to other viewpoints. For instance, in Figure 5, it is not shown whether the night scene's appearance is correctly maintained from different perspectives.

**Questions:**

The method mentioned in the text will have different memory overheads and training times depending on the number of images in different scenarios. Can you provide data on the GPU memory overhead and training times specific to each scenario?

**Limitations:**

The paper mentions that when there are too many occluded regions, this method cannot correctly fill in the areas after removing the occluders, and it may require the help of a diffusion model to resolve this. On the other hand, if this method needs to be optimized with complete ground truth for each new environment (occluders, illuminations, and weather), it can be said to have no generalization ability.
I will consider raising my score if the author can provide experiments with half-image test-time optimization.

---

> ### Author Rebuttal · Authors · 2024-08-06
>
> We thank the reviewer for the constructive feedback. We appreciate that you find that  “the performance in removing occluders is impressive”, and would like to raise the score if we can justify half-image test-time optimization. We will address all comments below and will adjust the paper accordingly.
>
> **W1: Novelty**
>
> We consider SWAG (published on arXiv on March 15th, latest revision: April 5th) as concurrent work and have developed our approach independent of the paper. While SWAG also used appearance embeddings, there are important technical differences.
>
> Similar to our approach, the authors of SWAG were motivated by Urban Radiance Fields to try predicting affine transformation parameters for the SH coefficients to model changes in appearance. The poor performance of this approach (see Tab. 8 in the SWAG paper) led the authors to conclude that “affine color transformations cannot model all appearance changes” and motivated the SWAG approach, which uses a feature grid with an MLP for color prediction. This MLP has to be evaluated for all Gaussians for each rendered view. As a result, SWAG has “10 times longer inference time per frame [5] compared to 3DGS”. Our approach shows that the conclusion drawn by the authors of SWAG regarding the expressiveness of affine transformations is wrong. In addition, our approach is both significantly faster to render and achieves better results (see PDF). We believe that this is interesting to the community.
>
> While our approach for handling transient objects is inspired by NeRF On-the-go, Fig. 3 in the main paper shows that the original formulation from NeRF On-the-go is not directly applicable to our problem setting. This is due to the formulation not being robust to appearance changes (which are not considered in NeRF on-the-go), see L186-195. We believe that showing how to adapt this approach to the case of changing appearance is interesting for the community.
>
>
> **W2: Fairness of the comparison**
>
> Thank you for raising this concern. For Photo Tourism, we actually follow the NeRF-W evaluation protocol (use half of the image for test-time optimization and only evaluate on the other half), but did not make this clear in the paper. For the NeRF On-the-go dataset, there is no test-time optimization needed. We will clarify this in the paper.
> Looking at the source code of GS-W (released after the NeurIPS submission deadline), we noticed that GS-W actually computes appearance information from the full test image. This indeed significantly boosts their performance. When using the NeRF-W protocol, our approach clearly outperforms GS-W. See the PDF for details.
>
>
> **W3: Appearance embedding applied to other viewpoints**
>
> We agree showing the consistency of the appearance modeling would benefit the paper. Unfortunately, we did not manage to complete the video in time. For the rebuttal, we include images sampled from our video in the PDF. As can be seen, the scene’s appearance is correctly maintained under viewpoint changes.
>
>
> **Question: GPU memory and training time per scene**
>
> Below, we provide the peek GPU memory and training times for individual scenes:
> | Scene | Num. Images | GPU Memory (GB) | Training Time |
> |--------------------|-------------|-----------------|---------------|
> | Trevi Fountain | 1689 | 39.2 | 10h 7m |
> | Brandenburg Gate | 763 | 8.0 | 6h 40m |
> | Sacre Coeur | 830 | 8.9 | 6h 2m |
>
> **Limitations: Incapable of recovering with too many occlusions**
>
> We wanted to express the following: If there are not enough observations of a part of the scene, e.g., because it is occluded in nearly all training images, our approach will struggle to correctly reconstruct the region, even if it filters out all occluders. The same limitation applies to other works in this field, e.g., NeRF On-the-go, GS-W, SWAG, NeRF-W, etc.: It is only possible to reconstruct areas that have been sufficiently observed. Still, as shown in our experiments, our approach works well without access to complete ground truth (we estimate occlusion masks during training rather than using ground truth masks) in realistic conditions. We will clarify this in the paper.

---

> > ### Comment · Reviewer_BAXf · 2024-08-10
> >
> > Thank you for your rebuttal. My primary concern has been adequately addressed through the additional experiments and detailed explanations you have provided. While I still have reservations regarding the novelty of this work, I do not think this significantly detracts from its overall contribution.  I will raise my rating to borderline or weak accept.

---

> > > ### Author Response · Authors · 2024-08-12
> > >
> > > Thank you for the response and we are pleased to see that our rebuttal has addressed the your concerns. We kindly ask you to raise the recommendation rating to match the accept rating. Thanks!

---

### Official Review · Reviewer_bi9Y · 2024-07-09

**Soundness:** 3
**Presentation:** 3
**Contribution:** 3
**Rating:** 7
**Confidence:** 5

**Summary:**

The authors present WildGaussians, an innovative method designed to address occlusions and appearance changes using 3DGS. By utilizing robust DINO features and incorporating an appearance modeling module into 3DGS, their approach achieves state-of-the-art performance. WildGaussians not only matches the real-time rendering speed of traditional 3DGS but also outperforms both 3DGS and NeRF baselines when dealing with in-the-wild data. This is accomplished within a straightforward architectural framework, making the method both efficient and effective. The results demonstrate that WildGaussians can handle complex scenarios involving dynamic objects and varying appearances, setting a new benchmark for real-time 3D rendering and modeling.

**Strengths:**

The overall approach of this paper is commendable, with detailed methods and experiments. Specifically: (1) By modeling Gaussian embeddings and image-specific embeddings to capture scene variations, it effectively represents both local and global perspectives; (2) By comparing the DINO features of rendered images and ground truth images, it reasonably models uncertainty and extracts masks, ensuring stable training for 3DGS.

**Weaknesses:**

The results section of the paper is relatively complete overall, but the results mainly focus on the removal of small-scale dynamic objects (Fig. 4). Theoretically, using DINO features can address large-scale occlusions. The authors could add some visual examples to illustrate this. Additionally, are the PSNR values in Tables 1/2/3 calculated on the images after masking? Additionally, could you compare the results with the latest SWAG [5] and GS-W [42], and then discuss the similarities and differences with them?

Typos: L145, it is possible pre-compute -> it is possible to ...; L102, handle occludersduring -> handle occluders during

**Questions:**

As discussed above.

**Limitations:**

Yes.

---

> ### Author Rebuttal · Authors · 2024-08-06
>
> Thank you very much for the positive and constructive feedback! We highly appreciate that you find our approach “commendable, with detailed methods and experiments” and that the paper “demonstrate[s] that WildGaussians can handle complex scenarios involving dynamic objects and varying appearances, setting a new benchmark for real-time 3D rendering and modeling”. We will address all concerns below, will fix the typos, and will adjust the paper accordingly.
>
> **Large-scale dynamic objects**
>
> Indeed, our method should be able to handle very large dynamic objects. However, please note that some of the objects removed are quite large. As we report in Table 3, for some scenes the occluders occupy on average 26% of the entire image. Moreover, in some scenes in the NeRF On-the-go dataset, occluders can even be buses or trams, which we consider “large occluders”. We will add examples to the paper. For the dataset visualization, please refer to the website of NeRF On-the-go.
>
>
> **Is PSNR computed after masking?**
>
> No masks were used for the PSNR computation (nor for the other metrics). The test images of both Photo Tourism and NeRF On-the-go do not have occlusions. We will clarify this in the paper.
>
>
> **Comparisons to SWAG and GS-W**
>
> We consider both SWAG (published on arXiv on March 15th, latest revision: April 5th) and GS-W (published on arXiv on March 23, latest revision: July 14th) as concurrent work. We developed our work independently from them. All three approaches (SWAG, GS-W, ours)  consider the same problem (modeling scenes with changing appearances from images containing occluders) and closely follow NeRF-based approaches proposed for the problem: they model appearance per Gaussian using appearance features stored per Gaussian and handle dynamic objects/occluders during training. They differ in the way both these parts are implemented:
> - To model appearance changes, **SWAG** uses a feature grid with an MLP for color prediction. This MLP has to be evaluated for all Gaussians for each rendered view. As a result, SWAG has “10 times longer inference time per frame [5] compared to 3DGS”. In contrast, our approach simply uses a shallow MLP to predict an affine transformation of the SH coefficients to obtain the final color. This approach is much more efficient and is “backward compatible” with 3DGS (see L144-147 in the paper), i.e., for a fixed target appearance it is not necessary to evaluate an MLP during rendering. To handle transient objects, SWAG introduces a trainable, image-dependent occupancy term per Gaussian. In contrast, our approach uses pre-trained DINO features to predict which training image regions contain static scene parts respectively dynamic objects.
> - To model appearance changes, **GS-W** combines per-Gaussian appearance features with features selected (per Gaussian) from a reference image to predict colors via MLPs. Similar to our approach, this approach to appearance modeling is significantly more efficient than the one used by SWAG. To model transient objects, GS-W trains a Unet-based model to predict a visibility map from a given training image, while our approach predicts uncertainties by comparing features from rendered and actual images.
> As can be seen, all three approaches differ significantly in the way they implement both stages. Additionally, our approach outperforms both SWAG and GS-W. We will extend our discussion in L86-91 to more clearly highlight the differences and add the results for SWAG and GS-W to the paper.
>
> The attached PDF compares our approach to both SWAG and GS-W. Please note that the results reported for GS-W differ from those reported by the authors of GS-W. The GS-W code, released about a month ago (i.e., after the NeurIPS submission deadline), uses the full test images for computing appearance information. The results reported for GS-W in the PDF were obtained by adjusting the code to follow the common test protocol used by all other approaches (including ours), which use one half of each test image to compute appearance information and the other half for evaluation. As can be seen, our approach outperforms both SWAG and GS-W. We will add the results to the paper.

---

> > ### Comment · Reviewer_bi9Y · 2024-08-12
> >
> > Thank you for the detailed rebuttal. Your responses have already addressed my concerns. This is a nice work, and I will keep my rating as accept.

---

> > > ### Author Response · Authors · 2024-08-13
> > >
> > > Thank you so much for your positive feedback on our rebuttal. We are truly glad that our responses have successfully addressed your concerns. Given your positive assessment, we are wondering if you could consider raising your rating from the current "borderline accept" to "weak accept" or higher?
> > >
> > > Thank you again for your time and consideration. We truly appreciate your valuable input throughout this review process.

---

> ### Comment · Reviewer_bi9Y · 2024-08-13
>
> I have raised my score to accept. Thanks for your efforts during the rebuttal and discussions. I hope the authors can include the comparisons and discussions in the revision to make the paper more solid and convincing.

---

> > ### Author Response · Authors · 2024-08-13
> >
> > We sincerely thank you for raising the score to 7. We greatly appreciate all reviewers' constructive feedback and fully agree that the revisions you suggested will enhance the quality of our paper. As promised, we will incorporate all of these improvements in the updated version.

---

### Official Review · Reviewer_5zzk · 2024-07-14

**Soundness:** 3
**Presentation:** 3
**Contribution:** 3
**Rating:** 6
**Confidence:** 3

**Summary:**

The authors are proposing WildGaussians, an approach based on 3D Gaussian Splatting (3DGS), that tries to address its robustness issues, specifically to significant appearance changes due to varying illumination or occlusions and dynamic objects.

First, explicit appearance modeling is introduced into 3DGS by using a small MLP to predict an affine color transform of the predicated color from the base color, image-specific appearance embeddings and per-Gaussian appearance embeddings.

Second, DINOv2 features, known to be robust to significant appearance changes, are leveraged to deal with occlusions: DINOv2 patch-level features are computed on training and predicted images, upscaled and binarized into a mask directly plugged into the optimization loss to mask out areas of low certainty.

The specific contributions of this submission are:
- extending 3DGS to support appearance modeling via additional embeddings combined with a tone-mapping MLP,
- extending 3DGS to be robust against appearance changes by using the similarity of DINOv2 features between training and predicted images to mask out uncertain regions,
- qualitative and quantitative evaluations of the combined changes on NeRF On-the-go and Photo Tourism datasets with comparison against relevant baselines (3DGS and NeRF On-the-go the NeRF On-the-go dataset or 3DGS, NeRF, NeRF-W-re, Ha-NeRF, K-Planes and RefinedFields on the Photo Tourism dataset).

**Strengths:**

- The submission is highly relevant to the research community since it deals wit improving the robustness of an influential technique on in-the-wild datasets. The proposed contributions are clear and simple extensions to 3DGS each addressing targeted robustness gaps and their presentation is solid and easy to follow thanks to this split between appearance and uncertainty modeling.

- The presented results are also solid. WildGaussians significantly outperforms baselines on NeRF On-the-go datase in Table 1 (except on low occlusions) and outperforms relevant baselines on Photo Tourism. The qualitative comparison of Figure 4 and 5 with selected artifacts in baselines help convincingly demonstrate the robustness of WildGaussians to occlusions and illumination changes.

- The (extended) ablations study (from the supplementary material) are extensive and cover the expected incremental changes and variants.

**Weaknesses:**

- The proposed changes to 3DGS seem of limited novelty:
  - appearance embeddings combined with an MLP to produce an affine mapping of colors is heavily inspired by NeRF derivatives like Urban radiance fields (with however some adjustments, such as per-Gaussian appearance embeddings and the required custom initialization), and
  - leveraging DINOv2 features to build an uncertainty mask is similar to NeRF on-the-go (CVPR'24 so quite recent though).
Applying variants of previous contributions to 3DGS is thus quite incremental.

- Some reference (and comparison) to related work appears to be missing: [Robust Gaussian Splatting](https://arxiv.org/abs/2404.04211) (April 2024). Note there are also several relevant concurrent works that have since appeared (but which would be unreasonable to call out as a weakness).

- The justification to introduce a binary uncertainty mask (over uncertainty weights) could be improved with more illustration of the problems encountered as well as the specific choice of threshold.

- Some minor typos to correct:
  - l.102 occludersduring -> occluders during
  - l.204: ocupacity -> apacity
  - l.510: borader -> broader

**Questions:**

- Why are the considered baselines different depending on the dataset? Only 3DGS is used on both the NeRF On-the-go and Photo Tourism datasets.

- Any additional insights on why WildGaussians fares worse with low occlusion in Table 1 (and how to mitigate this)? It seems the authors believe 3DGS is inherentily robust to low occlusion thanks to its initialization from an SfM point cloud.

---

> ### Author Rebuttal · Authors · 2024-08-06
>
> Thank you very much for the positive and constructive feedback! We really appreciate that you consider our work “highly-relevant to the research community”, and that our “contributions are clear and simple” while our method “significantly outperforms baselines”. We will address all concerns below, will fix the typos, and will adjust the paper accordingly.
>
> **W1: Limited novelty**
>
> While we are indeed inspired by both Urban Radiance Fields (URF) and NeRF on-the-go, there are important technical differences in applying these ideas in the context of 3DGS “in the wild”. We believe that they are interesting enough for the community to warrant acceptance.
> - **Differences to URF**: In order to handle exposure changes, URF only models a global affine transformation per image. In contrast, our per-Gaussian embedding enables handling local appearance changes in various parts of an image. The entry “w/o Gaussian embeddings [24]” in Tab. 4 corresponds to using the approach from URF. As can be seen, not using the per-Gaussian embeddings significantly reduces performance. The authors of SWAG report a URF-inspired approach for the same problem that performs significantly worse than their SWAG method (this observation is used to motivate the significantly more expensive SWAG approach, see our reply to reviewer nPTW for details). They conclude that “affine color transformations cannot model all appearance changes”. Our work disproves this statement as our method outperforms SWAG, as shown in the attached PDF. We believe that this is interesting to the community.
> - **Differences to NeRF On-the-go**: While our approach for handling transient objects is clearly inspired by NeRF on-the-go, Fig. 3 shows that the original formulation from NeRF on-the-go is not directly applicable to our problem setting. This is due to the formulation not being robust to appearance changes (which are not considered in NeRF on-the-go), see L186-195. We believe that showing how to adapt this approach to the case of changing appearance is interesting for the community.
>
> **W2: Missing related work**
>
> We will add Robust Gaussian Splatting and other recent concurrent works (SWAG, GS-W, etc) to the related work section, as well as comparisons to relevant methods (see the results shown in the attached PDF).
>
> **W3: More illustrations for the justification of a binary uncertainty mask**
> Thanks for the great suggestions. We promise to add more illustrations and the choices for threshold.
>
> **Q1: Why are baselines different**
>
> Photo Tourism and NeRF on-the-go pose different challenges (strong appearance changes and moderate occlusions vs. limited appearance changes and strong occlusions). Consequently, methods developed for one scenario are typically not evaluated in the other and vice versa (in addition, NeRF on-the-go was released only very recently). We use both datasets to showcase the robustness of our approach to both appearance changes and strong occlusions. For the rebuttal, we ran multiple baselines on both datasets (see PDF), leading to more shared baselines.
>
>
> **Q2: Why WildGaussians fares worse with low occlusion in Table 1, and how to mitigate it?**
>
> While NeRF on-the-go outperforms WildGaussians for low occlusions, the difference is marginal for PSNR (20.63 vs. 20.62) and SSIM (0.661 vs. 0.658). Without appearance modeling, WildGaussians outperforms NeRF on-the-go in both metrics (see Tab. 3). WildGaussians’ approach to handling transient objects is designed to be robust to appearance changes, and this added level of robustness compared to NeRF on-the-go seems to hurt the LPIPS performance of WildGaussians. The fact that 3DGS, Gaussian Occupancy Fields, and Mip-Splatting all achieve a similar LPIPS as NeRF on-the-go, suggests that occlusion handling is not very important for such a low level of occlusion.
>
> A potential way to mitigate performance degradation is to try to automatically detect which components (appearance modeling, handling transient objects) are necessary for a given scene.

---

> > ### Comment · Reviewer_5zzk · 2024-08-12
> >
> > I have read the rebuttal from the authors and the other reviews (as well as replies from the authors). I would like to thank the authors for their answers and appreciate their thoroughness in doing so. I also believe (most of) my concerns have been somewhat addressed, so I am still proposing to accept this submission.

---

> > > ### Author Response · Authors · 2024-08-13
> > >
> > > Thank you a lot for keeping the rating as acceptance, and we are glad that we have resolved your concerns. In case there are further concerns, we are happy to address them.

---

### Official Review · Reviewer_nPTW · 2024-07-16

**Soundness:** 3
**Presentation:** 3
**Contribution:** 2
**Rating:** 5
**Confidence:** 5

**Summary:**

The author proposes an improvement strategy for reconstructing 3D scenes from in-the-wild data based on the latest 3DGS method, primarily addressing occlusion and appearance changes. The main improvements are as follows:
1.Appearance Encoding with MLP: Introduce a Multi-Layer Perceptron (MLP) to encode the appearance of images. This involves two trainable embeddings as inputs: per-image embedding, per-Gaussian embedding, and base color (SH=0). The output consists of color transformation parameters, γ and β.
2.Uncertainty Modeling: Introduce uncertainty modeling to mask occlusions and dynamic objects. The author uses DINOv2 to calculate the feature similarity between the predicted image and the training image, thereby
guiding 3DGS to avoid reconstructing occlusions.
The motivation of the article is relatively clear, the structure is complete, the content is substantial, and the writing is easy to understand with minimal writing issues. The experimental part is also quite comprehensive, which to some extent demonstrates the effectiveness of the proposed improvement method. However, the submission did not include a video, which reduced the persuasiveness of the paper's results in the task of NVS and view transitions.

**Strengths:**

1. The problem that the authors attempt to solve is indeed a real one, and there is a significant demand in the industry for reconstructing 3D scenes from in-the-wild data.
2. The idea of leveraging DINO features for jointly optimizing occlusion is sound and interesting.
3. The paper has a complete structure, with clear language expression and minimal writing issues. The content is well-organized and easy to understand.

**Weaknesses:**

1. The two improvement ideas proposed in the article (MLP color mapping and uncertainty model) are not uncommon, especially in NeRF research. Although 3DGS is still in its early stages, many works such as SWAG, GS-W, and Scaffold-GS also mention appearance encoding and the introduction of pre-trained models. Therefore, I do not see any significant differences between this work and these existing works. Additionally, in the calculation of feature similarity, a pre-trained model was used for fine-tuning, making this work seem more like a combination of existing methods (A+B stitching) rather than presenting original contributions or theoretical derivations.
2. In Lines 126-128, the author mentions that the latest methods (e.g., mip-splatting, Absgs, Gaussian Opacity Fields) are introduced to improve the original 3DGS, indicating that the research in this article is based on a stronger baseline than the original 3DGS. However, in the experimental comparison, it is only compared with the original 3DGS. As far as I know, the methods introduced also aim to address issues with artifacts and floaters. Therefore, it is difficult to determine how much the performance enhancements are due to the improvement strategies proposed in this paper. A comparison with the stronger baseline should be added in Tables 1, 2, and 3.
3. I don't think the comparisons shown in Figures 4 and 5 are very effective. Although there is an improvement in simulating ambient light compared to the original 3DGS, I cannot agree with the statement in the article that "we can adeptly handle changes in appearance such as day-to-night transitions without sacrificing fine details." Clearly, in terms of stone carvings, wall details, and water surface textures, the original 3DGS appears sharper. Perhaps your improvement sacrifices some details but achieves better fitting for appearance changes and smoother view transitions. Unfortunately, you did not submit a video to support this.
4.Some typos：
-Figure2 can be more clear,you can add some legends to indicate the gradient flow. And what does "affi" stands for?
-The Eq.(2) is wrong(incomplete).
-The superscript in Eq.(7) is incorrect:C~
-L.430,the output of MLP is (β,γ) as mentioned in L.134

**Questions:**

1. Related Work (Sec. 2)
In Line 87, you mention two related works, SWAG and GS-W, but the results analysis section does not compare your method with theirs. I understand that they also use MLP to encode appearance. However, what is the most significant difference between your work and theirs? This distinction does not seem very clear.
2. Method (Sec. 3)
	1)Regarding Appearance Modeling, do you still use SH=3 to represent the color of Gaussians? Given that neural networks are already being used, why not directly predict the final color like Scaffold-GS? Instead, you predict color transformation parameters. Could you clarify this choice?
	2)In Line 155, you mention that "DSSIM and L1 are used for different purposes in our case." However, it is still unclear why one uses "image rasterized without appearance modeling" for calculation, while the other uses the correct appearance for calculation. You also state that DSSIM is more robust to appearance changes and focuses more on structure and perceptual similarity. Shouldn't it be possible to calculate DSSIM on the corrected image?
	3)Is Equation (11) the entire loss function term?
	4)In Line 220, you mention that "we project them to all training cameras, removing any points not visible from at least one camera." However, my understanding is that the pruning strategy of the original 3DGS should already prune out invisible points. Is this step critical?
3. Experiments (Sec. 4)
	1)In Table 1, why are there only two methods compared on this dataset, while more methods are compared on the Photo Tourism Dataset in Table 2?
	2)In Table 2, why is there such a significant difference in FPS between your method and 3DGS? I would expect your method to have a similar rendering speed to 3DGS. Is the difference due to the introduction of previous methods to 3DGS (Mip splitting, Absgs, Gaussian Opacity Fields)? These methods might compress and prune the number of Gaussian points, making it an unfair comparison to the original 3DGS.
	3)Ablation Study: Sky Handling. You have three improvements, but I haven't seen any ablation experiments on sky handling. Is its impact on the results considered less significant?
	4)Can you explain why adding appearance modeling to some datasets in Table 3 actually leads to a decrease in metrics? Based on all the experimental results, can I understand it this way: the methods proposed in this paper are effective primarily under conditions of strong occlusion and significant appearance changes. For minor occlusion or appearance changes, using the methods in this paper may not necessarily yield improvements and could even decrease the performance metrics?

**Limitations:**

The authors have adequately addressed the limitations and potential negative societal impact of their work.

---

> ### Author Rebuttal · Authors · 2024-08-06
>
> Thank you very much for the positive and constructive feedback! Below, we address the concerns raised in the review. We will adjust the paper accordingly and fix the typos.
>
> **W1: The ideas in the paper are not uncommon, similar to SWAG, GS-W, and Scaffold-GS**
>
> To our understanding, Scaffold-GS (CVPR 2024) neither uses appearance encoding to model appearance changes nor pre-trained models, and it is not evaluated on datasets with significant appearance changes or moving occluders.
>
> We consider SWAG and GS-W as concurrent work. Our approach, developed independently, addresses the same problem (modeling scenes with changing appearances and occluders) and follows NeRF-based methods but differs in implementation:
> - **SWAG**: Uses a feature grid with an MLP for color prediction, leading to slower inference. We use a shallow MLP for an affine transformation of SH coefficients, making it more efficient. For transient objects, SWAG uses a trainable, image-dependent occupancy term per Gaussian, but we use pre-trained DINO features to predict regions with dynamic objects.
> - **GS-W**: Combines per-Gaussian appearance features with features from a reference image via MLPs. It uses a Unet model to predict visibility maps, but we predict uncertainties by comparing features from rendered and actual images.
>
> All three approaches differ significantly in implementation. Additionally, our approach outperforms both SWAG and GS-W (cf. PDF). We will extend discussion, and add results for SWAG and GS-W.
>
> **W1: This work seem like a combination of existing methods**
>
> As detailed above, our approach differs significantly from existing approaches in technical details.
> - **Appearance modeling:** SWAG shows that our approach is non-trivial: it reports that authors tried to predict an affine transformation of the SH coefficients, but observed significantly worse results than with their approach (cf. Tab. 8 in 2403.10427v2). They conclude that “affine color transformations cannot model all appearance changes”. We show that this conclusion is false, as we outperform SWAG.
> - **Transient objects:** Ours is the first work in the context of 3DGS that uses a pre-trained foundation model for uncertainty modeling.
>
> **W2: Comparisons over stronger baselines**
>
> Similar to SWAG and GS-W, we compare to standard 3DGS in Tab. 1, 2. Tab. 4 contains a comparison with a stronger baseline: *w/o appearance&uncertainty* corresponds to Mip-Splatting+AbsGaussian(GOF) + our sky modeling. This baseline is not substantially better than 3DGS on Photo Tourism (PSNR: 18.48 vs. 18.13).
> We also add a comparison on NeRF On-the-go dataset in the PDF, where we clearly outperforms both the baselines and GS-W, especially with lots of occlusions. We will add the results and baselines to Tab. 1, 2.
>
> **W3: Disagreement with the claim to “handle changes in day-to-night transitions without sacrificing fine details”**
>
> Indeed, being able to handle appearance changes seems to come at the price of a (slight) loss in detail. We will soften the claim to “Compared to 3DGS, we can adeptly handle changes in appearance such as day-to-night transitions at the cost of a slight blurring in fine details.”
>
> **Q1: Comparison to SWAG and GS-W**
>
> GS-W only released code about a month ago and SWAG has not released code. Looking at the code of GS-W, it uses the full test image for computing the appearance (see our reply to BAXf for details), and fixing the code to use NeRF-W evaluation protocol significantly reduces performance (see PDF). Our approach outperforms both SWAG and GS-W.
>
> **Q2: Why not predicting color directly like Scaffold-GS?**
>
> We still use SH coefficients so that after fixing the appearance embedding, we can bake our representation back to 3DGS for better portability and faster runtime. In our experience, predicting offsets from a base color stored in Gaussians leads to more stable training.
>
> **Q2: Use of DSSIM and L1**
>
> We follow VastGaussian and use L1 to ground the appearance (taking the final color into account) and DSSIM to ground the structure. Using the renderings without appearance modeling, which increases the performance slightly in our experiments. We will clarify this in the final version.
>
> **Q2: Eq. (11) entire loss function?**
>
> The loss is sum of Eq. 11 and Eq 9. We will clarify this.
>
> **Q2: When initializing sky Gaussians, why remove points not visible in any camera when 3DGS will prune them automatically?**
>
> We remove these points as they will never be seen in any view and, therefore, there is no point adding them and slowing down the training (3DGS will never prune points not visible in any camera).
>
> **Q3: Only two methods compared on NeRF On-the-go dataset**
>
> The dataset was released right before the submission deadline, and we were only able to compare to 3DGS and NeRF On-the-go in time. Above, we show results for GS-W, Mip-Splatting, and Gaussian Opacity Fields. We will include the results in Tab. 1.
>
> **Q3: Unfair to compare with 3DGS when you use Mip-Splatting and AbsGaussian (GOF)**
>
> In Table 4, we also report the numbers on our model with app. modeling and uncertainty modeling disabled. This is essentially Mip-Splatting+AbsGaussian(GOF) with our sky modeling.
>
> **Q3: Why is 3DGS slower than WildGaussians?**
>
> 3DGS cannot explain the transient objects and it grows Gaussians to account for that. The excessive Gaussians (floaters) slow it down. We will make this clearer in the final version.
>
> **Q3: Ablation study for sky handling**
>
> We will add an ablation study to Table 4. Unfortunately, we did not have time to do this for the rebuttal.
>
> **Q3: Why adding appearance modeling to datasets without appearance changes decreases the performance?**
>
> Indeed, enabling appearance modeling for datasets with no appearance changes reduces performance slightly. The same happens with NeRFs, e.g., see Mip-NeRF 360 paper. The added (unnecessary) degrees of freedom, in our case appearance embeddings, make the optimization problem more difficult.

---

> > ### Comment · Reviewer_nPTW · 2024-08-13
> >
> > After reading all the review and the rebuttal, I would like to keep my rating since the rebuttal addressed most of my concerns.

---

### Author Rebuttal · Authors · 2024-08-06

**Global Response**

We thank all reviewers for their constructive comments. Please check the attached one-page PDF for more numerical and visual results on:

* **Figure 1**: Multiview consistency for single app. embedding
* **Table 1**: SWAG and GS-W comparison on Photo Tourism
* **Table 2**: Additional baselines comparison on NeRF On-the-go

---

### Author Response · Authors · 2024-08-12

We thank the reviewers and are pleased to see that our rebuttal has addressed the reviewers’ concerns.
We are happy to provide further details, discussions  etc. If there is anything we can do in that regard, please let us know.

---

### Decision · Program_Chairs · 2024-09-25

**Decision:**

Accept (poster)

**Comment:**

All reviewers agreed to accept the paper. The paper originally got BA, WA, BA, and R, and, after the rebuttal period, the last two reviewers raised the ratings, leading to final ratings of BA, WA, A, and WA. Reviewers recognized that the paper has clear contributions to 3DGS for in-the-wild image inputs, particularly in appearance modeling and uncertainty modeling. However, reviewers are also concerned whether the improvement would not be significant enough. After the authors' rebuttal and discussion period, reviewers mentioned that their concerns have been mostly resolved, and concluded that the paper's shape is above the bar for NeurIPS.  The AC also supports accepting the paper and strongly suggests the authors to follow the reviewers’ feedback in the camera-ready version.